# Assessment of Bandaged Burn Wounds Using Porcine Skin and Millimetric Radiometry

**DOI:** 10.3390/s19132950

**Published:** 2019-07-04

**Authors:** Amani Yousef Owda, Neil Salmon, Sergiy Shylo, Majdi Owda

**Affiliations:** 1School of Electrical and Electronic Engineering, University of Manchester, Sackville Street Building, Manchester M13 9PL, UK; 2School of Engineering, Manchester Metropolitan University, Chester Street, Manchester M1 5GD, UK; 3Usikov Institute of Radiophysics and Electronics National Academy of Sciences of Ukraine, 61085 Kharkov, Ukraine; 4School of Computing, Mathematics and Digital Technology, Manchester Metropolitan University, Chester Street, Manchester M1 5GD, UK

**Keywords:** dressing materials, burn wound, radiometry, millimeter-wave, passive imaging

## Abstract

This paper describes the experimental setup and measurements of the emissivity of porcine skin samples over the band of 80–100 GHz. Measurements were conducted on samples with and without dressing materials and before and after the application of localized heat treatments. Experimental measurements indicate that the differences in the mean emissivity values between unburned skin and burned damaged skin was up to ~0.28, with an experimental measurement uncertainty of ±0.005. Measured differences in the mean emissivity values between unburned and burn damaged skin increases with the depth of the burn, indicating a possible non-contact technique for assessing the degree of a burn. The mean emissivity of the dressed burned skin was found to be slightly higher than the undressed burned skin, typically ~0.01 to ~0.02 higher. This indicates that the signature of the burn caused by the application of localized heat treatments is observable through dressing materials. These findings reveal that radiometry, as a non-contact method, is capable of distinguishing between normal and burn-damaged skin under dressing materials without their often-painful removal. This indicates the potential of using millimeter wave (MMW) radiometry as a new type of medical diagnostic to monitor burn wounds.

## 1. Introduction

A burn is a thermal injury or damage caused by chemical, electrical current, or radiation (including that from the sun). In England and Wales, around 13,000 patients with burn injuries are admitted to burn services departments annually [1]. According to 2003–2011 statistics, males (females) accounted for 63% (37%) of those patients with a total in-hospital mortality of ~1.51% [1]. The severity of the burn can range from minor (erythema or redness) to extremely severe (full thickness). The severity is characterized by the extent of the skin affected, the depth of the damage, the age of the patient, the anatomical site, and the presence of disorders [1,2]. Burn is classified by depth into; first degree, second degree, and third degree [3] as illustrated in Figure 1.

Human skin is a self-healing organ that can go through complex dynamic processes to re-establish and repair the damaged areas. The skin might repair itself successfully, but if not disorders and mortality may result. Characteristics and features for partial and full thickness burns are summarized in Table 1 [6,7,8].

Burn wounds require treatment to prevent fluid and protein loss, the risk of infection, and hypothermia [9,10]. Wound healing is a complex dynamic process of tissue regeneration and growing that requires a good environment [11]. Dressing material is used in the treatments of burn wounds since (1) it absorbs the exudates, (2) it provides a barrier against bacteria and organisms that cause infection, that would otherwise prolong the healing period, (3) it maintains an appropriate temperature and moisture level, (4) it allows gas exchange between the environment and the wound site, and (5) it enhances the blood flow.

Visual inspection is the current protocol for monitoring the burn wound healing progress [3,12,13]. This protocol gives an excellence indicator about the state of the wound and the healing process, and more importantly, it can detect signs of infections such as exudates, redness, swelling, heat, functionality of the infected part, and pus draining [14]. However, visual inspection requires the removal of dressing layers. This practice consumes time, money, and it is definitely uncomfortable and painful to the patient, especially in the cases where the dressing materials become moistened and adhere to the wound bed [11,15].

Although many signs can help the experienced burn surgeon to identify the depth of the burn, making a distinction between the second and the third degree burns visually is a difficult task. The accuracy of this assessment is around 64% to 76% [16,17] and it can fall to 50% if the burn is assessed by inexperienced surgeons [16,18]. False assessments of burn depth prolong the time required for a burn wound to heal and cause the patient sometimes to face unnecessary surgery [16]. Therefore, many techniques have been developed for assessing the wound healing progress and the degree of the burn such as optical coherence tomography [2,19,20,21], thermography (including the active dynamics thermography and infrared thermography) [22,23,24,25], and ultrasound (including pulse-echo ultrasound, pulse-wave Doppler ultrasound, and laser Doppler ultrasound) [26,27,28,29]. Although some of the results obtained from these technologies are promising, none of these techniques can monitor the wound healing progress and the degree of the burn without the removal of the dressing materials. A technique that could penetrate dressings and identify the healing status of the burn wounds is of great interest to patients, healthcare professionals, the National Health Service, and the private healthcare industry.

Recently, Gao and Zoughi [6,30] suggested that MMW reflectometry—i.e., active radiation—could be used as a non-invasive in-contact technique to distinguish between unburned and burned skin having different degrees of burn injuries. This paper investigates the feasibility of using radiometry—i.e., passive radiation—as a non-contact technique to distinguish between normal and burn-damaged skin signatures with and without the presence of dressing materials. The technique presented in this paper—i.e., radiometry—can measure the emission of the MMW radiation in tens of seconds by using a non-contact sensor with high precision and it does not involve artificial sources of radiation, only naturally present emission from the environment [15,31].

A motivation for measuring porcine skin in this research is that it has structural and functional similarities to the human skin [9,32]. Porcine and human skin has similar percentages of collagen and elastic fibers in their extracellular matrix [9]. The thickness of the epidermis layer for porcine skin varies depending on the location in the range of 30–140 μm, whereas the thickness of the epidermis layer for the human skin varies in the range of 50–120 μm [9,33]. In addition, human and porcine skin has resemblances in elasticity [9], thermal sensitivity [34], keratin types [35], vascularization [9], and stratum corneum thickness [35].

In this research, 15 porcine skin samples were purchased from an abattoir and the measurements were conducted on these for a time of up to no longer than four hours after the slaughter. The samples were taken from pigs having ages ranging from six to eight months and average weights from 55 kg to 60 kg. The samples were taken from the back region of nine healthy animals. This region is chosen since it is free from hair follicle and sweat glands. The study was approved by the ethics committee of Manchester Metropolitan University under ethics reference no. SE1617114C.

## 2. Materials and Methods

### 2.1. Three Layer Model for Dressed Burn Wound

A three-layer model has been constructed and developed to determine the emissivity of the dressed burn-damaged skin; in this model, the first layer is a semi-infinite layer of air, the second layer is a finite thickness dressing materials, and the third layer is a semi-infinite layer of skin, as illustrated in Figure 2.

In the model, radiation from the surrounding environment, having a radiation temperature *T_o_*, illuminates the dressing materials, whilst radiometric emission from the skin having a physical temperature *T_S_* illuminates the reverse side. As the dressing material, at a physical temperature *T_D_* has finite absorption, these are also radiating. The radiation temperature of the sample (dressed burn-damaged skin) as measured by the radiometer can, therefore, be expressed [36] as
(1)$$T_{b}=T_{o}R\left(1-b\right)+T{’}_{N}Rb+T_{D}A+T_{S}\eta ,$$
where *T_N_*, is the receiver noise temperature of the radiometer, *R* is the reflectance of the dressed skin, *A* is the absorptance of the dressing layer, and *η* is the emissivity of the skin. The parameter *b* is the fraction of the total radiation blocked from being reflected from the sample by the antenna, the value of *b* depends upon the antenna and its proximity to the sample. Law of conservation of energy applied on the system gives [37]
(2)1=R+A+η,

A wave-guide circulator was placed between the horn antenna and the receiver and this makes, *T_N_* = *T_o_*. For a very low loss dressing materials, *A* ≈ 0, and this means the radiation temperature of the sample in Equation (1) can be expressed in terms of the sample emissivity. The sample thermodynamic (or physical) temperature and the background illumination radiation temperature then becomes
(3)$$T_{b}=\left(1-\eta \right)T_{o}+T_{s}\eta ,$$

A radiometer is an instrument that measures the thermal (Planck) radiation. For radiation frequencies below the mid-infrared band the intensity of the emission is directly proportional to the temperature of the object, thus enabling regions of the image to be calibrated in degrees Kelvin [38]. Calibration of the radiometer is done using two stable blackbody radiator sources, one at a low temperature *T_C_* and the other at a high temperature *T_H_* [37,39,40]. This process calibrates the receiver responsivity and the receiver noise temperature, the assumption being that the system responsivity is linear. If the measurements are done indoors in an anechoic environment (where there is no radiometric emission from people or lower emissions from outdoors), the low temperature calibration source is a foam absorber at ambient temperature *T_C_*. Under these circumstances, the output voltage when measuring the low temperature calibration source V_C_ becomes [37]
(4)$$V_{C}=\;\alpha \left(T_{C}+T_{N}\right),$$
and when measuring the high temperature calibration source *V_H_*, the high temperature calibration source is a piece of foam absorber heated and stabilised at 54.0 °C, it is
(5)$$V_{H}=\;\alpha \left(T_{H}+T_{N}\right),$$
From this calibration procedure, the receiver responsivity is
(6)$$\alpha =\frac{\left(V_{H}-V_{C}\right)}{\left(T_{H}-T_{C}\right)},$$
and the output voltage for the sample V_S_ is
(7)$$V_{s}=\alpha \left(T_{b}+T_{N}\right),$$

Using Equations (3), (6) and (7), and equating *T_o_* to *T_C_* as the cold load is a foam absorber at ambient temperature; the emissivity of the sample becomes
(8)$$\eta =\frac{\left(Vs-V_{C}\right)(T_{H}-T_{C})}{(V_{H}-V_{C})\;\left(T_{s}-T_{C}\right)},$$
where, *V_s_*, *V_H_*, and V_C_ are the voltage levels in volts for the sample, the hot and the cold calibration loads respectively, and *T_s_*, *T_H_*, and *T_C_* are their respective thermodynamic temperatures in Kelvin.

### 2.2. Experimental Work

The objectives of the experimental work are: (1) to develop an experimental setup for measuring the emissivity of porcine skin samples, (2) to assess the feasibility of using the 90 GHz calibrated radiometer as a non-contact technique to distinguish between normal and burn-damaged skin signatures with and without the presence of dressing materials, and (3) to assess the feasibility of using the 90 GHz calibrated radiometer to predict the degree of the burn (or the burn depth) with and without the presence of dressing materials.

To achieve these objectives, the experimental work presented in this paper is divided into two parts; (1) calibration measurements of the 90 GHz radiometer, and (2) emissivity measurements for porcine skin samples before and after the applications of localized heat treatments with and without the presence of dressing materials.

#### Experimental Description

A direct detection radiometer sensitive over the frequency band (80–100) GHz was used for measuring the emissivity of porcine skin samples. This frequency band was chosen as indications are that radiation at this band interacts with the top layers of the skin and as such is ideal for measuring the epidermis and the dermis layer of the skin. A motivation of using radiometry in this research is that the technique does not involve artificial sources of radiation, only naturally present emission from the environment. This means that there are no health perception issues associated with radiometry. The radiometer consisted of a W-band horn antenna connected directly to the millimeter wave monolithic integrated circuit (MMIC) receiver. The output of the receiver was connected through a coaxial cable to a digital voltmeter and through wires to a DC power supply as illustrated in Figure 3.

The MMIC receiver consisted of a two-stage low noise amplifier, zero bias diode detector, and buffer amplifier. The output voltage from the detector is directly proportional to the power of millimeter wave radiation and this is proportional to the radiation temperature given by Equation (1). The complete system except for an opening for the subject to be measured was enclosed in an anechoic region made by surrounding the detector and the horn antenna with surfaces of carbon loaded absorbing foam (type: anechoic pyramidal absorber and dimensions are; length = 1200 mm, and width = 2400 mm). The horn antenna (model number: AS4341, manufacturer: Atlan Tec RF) has a rectangular aperture (30 × 25 mm) and a nominal gain of 20 dBi over the frequency band 80–100 GHz. The horn antenna was moved by hand to measure emissions from the porcine skin samples in normal and burn damaged state with and without the presence of the dressing materials and also from the hot and the cold calibration sources as illustrated in Figure 4. The measurements were performed over a distance of 5.0 cm from the skin sample and calibration sources. This distance has been chosen as an optimal distance to minimize the chances of subjects accidentally touching and moving the measurement apparatus. A greater distance between the measured subject and the horn antenna would lead to measurements having poorer spatial resolution.

The carbon foam absorbers (type: Eccosorb AN-73, manufacturer: Laird) had a rectangular shape and dimensions (length = 170 mm, width = 150 mm, and thickness = 10 mm). These dimensions were chosen to fill the beam pattern of the horn antenna, thereby minimizing systematic uncertainties. The measured emissivity values of the foam absorbers are greater than 0.99 over the frequency band (80–100) GHz [41,42], thus they behave as good approximations to a black body emitter. Radiometers have the performance metric of noise temperature measured in Kelvin; the lower the figure, the more sensitive the system. The noise temperature of the radiometer was measured in the course of this research to be 430.5 K, which represents a good performance for this application. Calibration measurements were taken from 10 separate experiments and repeated 5–10 times and for each time they were found to be stable and consistent. This indicates that the radiometer had long-term measurement stability.

The amount of self-emission reflected back from subjects was investigated by placing a metal plate perpendicular to the beam at a distance of 1.0 cm from the horn antenna beam (this distance was chosen as it allows maximum reflected emission to be observed without touching the apparatus). The mean level of self-emission reflected back from the metal plate (100% reflective surface) was measured to be in the range of 294–295 K with a standard deviation of ±1.0 K. These results show that the radiation temperature from the metal plate is approximately the same as the ambient temperature, meaning there is no spurious emission from the radiometer to corrupt the measurements [43].

Fifteen fresh porcine skin samples were taken from the back regions of nine animals to be used in this research. In general, the samples have a rectangular shape and average dimensions were (length = 100 mm, width = 80 mm, and thickness = 5.0–10.0 mm); all samples were chosen to be free from the hair follicles. The samples were taken directly after the animal was slaughtered and before the skin was washed.

A digital hotplate (type: LED digital hotplate magnetic stirrer, manufacturer: SciQuip Ltd. Shropshire, UK) with a temperature range of 280 °C was located inside a polystyrene foam bucket and used to heat the samples and to stabilize the skin surface temperature to ~37 °C as illustrated in Figure 5a. This temperature is chosen since it is closed to the in vivo surface temperature of the porcine skin ~35 °C [32,44].

The heat control metal plate shown in Figure 5b consists of a temperature controller, thermocouple, and a square metal plate (50 × 50 mm). During the experimental work, the device was used to apply a contact burn after the plate was heated in the range of 100–140 °C and placed on the skin surface with a constant pressure for different periods of time ranging from 10 s to 180 s. These periods were chosen to achieve different degrees of burn and different burn depths. The degree of the burn was assessed based on the visual signs such as the color, the amount of damage, and swelling. Dressing materials (types: gauze burn dressings and light support bandages) were placed over the skin sample when this was required.

A standard thermocouple (model number: L812, manufacturer: Leaton) was used to measure the skin surface temperature of the samples. The temperature is indicated via a digital readout with a ±0.5 °C absolute measurement uncertainty and 0.1 °C step size. An infrared thermometer (model number: N85FR, manufacturer: Maplin) with a temperature range −50 °C to +550 °C and resolution of 0.1 °C was used to measure the temperatures of the skin and the calibration sources. The devices are cross-calibrated by measuring the temperature of the same source, so the relative uncertainty of the measurement is much smaller, typically less than 0.1 °C. A digital voltmeter (type: digital voltmeter, manufacturer: Keysight Technologies) with 0.1 mV step size was used to measure the voltage level of the thermal emission of the porcine skin samples. Error propagation through Equation (8) indicates that the uncertainty on the measured emissivity is ±0.005.

### 2.3. Methodologies Applied on Porcine Skin Samples

This section discusses different methodologies developed and applied on the porcine skin samples for measuring the emissivity of the skin before and after the applications of localized heat treatments with and without the presence of dressing materials.

#### 2.3.1. Methodology 1: Skin without Burns

The samples were located over a digital hotplate and left to be heated and stabilized to 37.0 °C. Then the calibrated radiometer of Figure 3 was used to measure the voltage level of the thermal emission emitted from the samples and a thermocouple and an infrared thermometer were used to measure the skin surface temperature. The measurements were repeated five times (to grade against the random errors and obtain a mean value) and processed using Equation (8).

Then dressing materials (six-layer Gauze burn bandage, and single-layer light support bandage) were placed separately over the samples and the emissivity measurements were conducted directly using the calibrated radiometer of Figure 3 and Equation (8).

#### 2.3.2. Methodology 2: Skin with Burns

The samples were located over the digital hotplate and left to be heated and stabilized to 37.0 °C. Then contact burns were applied using a heat control metal plate. The plate was heated to 140 °C and placed directly on the skin surface for a period of time ranging from 10 s to 180 s with a constant pressure. Then emissivity measurements for the burn-damaged skin were obtained and repeated five times using the calibrated radiometer of Figure 3 and Equation (8).

Then dressing materials were placed over the burn-damaged skin and the emissivity of the sample was measured using the calibrated radiometer of Figure 3 and Equation (8).

#### 2.3.3. Methodology 3: Skin with Different Burn Depths

The samples were located over the digital hotplate and left to be heated and stabilized to 37.0 °C. The emissivities of the samples were obtained using the calibrated radiometer. Then contact burns were applied using a heat control metal plate heated to 100 °C and placed directly on the skin surface for a different period of time; start from 10 s (first degree burn), then 60 s (second degree burn), and finally 120 s (third degree burns). These periods of time were chosen as they are sufficient to produce different burn depths as assessed experimentally using samples under tests. In addition, the degree of the burns was assessed based on visual signs illustrated in Table 1. The emissivities of the samples after each application of localized heat treatments were obtained using the calibrated radiometer of Figure 3 and Equation (8).

## 3. Results

This section presents emissivity measurements made on porcine skin samples over the frequency band 80 GHz to 100 GHz. The measurements were conducted on samples with and without dressing materials and before and after the application of localized heat treatments.

### 3.1. Porcine Skin Measurements without Burns

This section presents emissivity measurements made on porcine skin samples over the band 80 GHz to 100 GHz; the mean emissivity values of the porcine skin samples were measured experimentally using methodology 1. The measurements were made on four fresh samples taken from the same animal. The measurements were repeated five times on each sample. The mean and the standard deviation of the measurements are illustrated in Figure 6.

The measurements in Figure 6 indicate that the mean emissivity value of the samples A, B, C, and D is ~0.52 with experimental measurements uncertainty of ±0.005. The standard deviations of the samples were calculated to be in the range of ~0.01 to ~0.02. It is reasonable to obtain the same mean emissivity value for all the samples as they were taken from the back region of the same animal.

The capability of the 90 GHz calibrated radiometer to detect the signature of the porcine skin under dressing materials was investigated and measured using methodology 1. The methodology was applied on samples A, B, C, and D. The mean emissivity values of the samples were obtained before and after the dressing materials were placed as shown in Figure 7.

The measurements in Figure 7 indicate that the differences in the mean emissivity values between the undressed and the dressed samples are in the range of ~0.01 to ~0.02 for all measured samples. These results confirm that the signature of the skin is seen through the six-layer gauze burn bandage and the light support bandage as the mean emissivity values of the dressed samples are very close to that of undressed samples.

### 3.2. Porcine Skin Measurements with Burns

The signature of the burn-damaged skin after the application of localized heat treatments was measured experimentally using methodology 2. The methodology was applied on four samples X, Y, Z, and W. These samples were taken from the back region of the same animal. The measurements were conducted before and after the dressings were applied on the samples. The measurements were repeated five times and the mean emissivity values were obtained as illustrated in Figure 8.

The measurements in Figure 8 indicate that the differences in the mean emissivity values between the unburned and the burned skin are: 0.044, 0.084, 0.184, and 0.264 for samples X, Y, Z, and W respectively. These differences confirm that there is a clear signature for the burn that can be detected using radiometry. The measurements also show that the signature of the burn is observed through the six-layer gauze burn bandage and the light support bandage, as the mean emissivity values of the dressed burn are slightly higher than the undressed burn in the range of ~0.01 to ~0.02. The measurements also indicate that there is a direct proportionality between the mean emissivity values of the burn-damaged skin and the period of time for which the localized heat treatment was applied. It is believed that this relationship arises as burn depth increases with the exposure time to the heat source [44,45].

### 3.3. Porcine Skin Measurements with Different Burn Depth

The capability of the 90 GHz calibrated radiometer to distinguish between different burns depths was investigated using methodology 3. The methodology was applied on seven different samples taken from the back region of seven different animals. Different applications of heat treatments were applied on the same samples and the mean emissivity values of the samples were measured before and after different applications of heat treatments as shown in Figure 9, Figure 10 and Figure 11.

The measurements in Figure 9 indicate that the mean emissivity values for the four samples before the application of heat treatments are in the range of 0.52–0.55. It is reasonable to find differences in the mean emissivity value of the skin as the samples were taken from different animals in which each animal has a different skin thickness and different dielectric properties [46,47]. The measurements also reveal that biological tissue is responding similarly to the application of heat treatment as the mean emissivity values increase after each application of heat treatment for all samples as summarized in Table 2. However, the mean emissivity values for the burn-damaged skin are different from sample to sample and this is due to the resistivity and the conductivity of the skin, as that varies from animal to animal [46,47,48]. The measurements presented in Figure 9 confirm that radiometry can distinguish between different burn depths as it provides different mean emissivity values after each application of localized heat treatment.

Heat treatment alters the chemistry and the structure of the biological tissue and more importantly cauterizes parts of the tissue. This process produces exudates around the wound site. Although for ex vivo tissue, it is difficult to observe exudates with the absence of blood circulation. However, during the experimental work, one of the samples produces exudates and as a result, the effect of exudates around the burn wound has been measured and investigated as illustrated in Figure 10.

The measurements in Figure 10 indicate that the mean emissivity value of the burn-damaged skin with exudates (L1) is lower than the normal skin by ~0.09. Whereas, the mean emissivity values of the burn-damaged skin without exudates and with different burn depths (L2 and L3) are higher by ~0.07 and ~0.28, respectively.

Microwave and MMW radiation sensors are very sensitive to the variations in water content in the biological tissues. The application of localized heat treatments makes the emissivity of the skin higher as the water in the skin is gradually evaporated. This has been investigated and measured by applying extra heat treatments on two samples as illustrated in Figure 11.

The measurements in Figure 11 indicate that the mean emissivities of the burn-damaged skin are in the range from ~0.63 to ~0.8, this being higher than that of the unburned skin. This is likely to be due to the application of heat drying the skin out and damaging its ability to exudate. For example, samples m3 and f3 contain almost no water. Then further heat treatment applied on the sample will not change the emissivity of the sample as in m4 and f4. This behavior is observed experimentally in the measurements of the two samples presented in Figure 11.

## 4. Discussion

The measurements conducted on 15 porcine skin samples from the back regions of the animals over the band 80 GHz to 100 GHz indicate that the mean emissivity values of the porcine skin without burns are in the range of 0.52 to 0.55. These values are within the range of the human skin emissivity measurements of the palm of the hand region [49]. These results confirm that porcine skin is a good phantom model for the human skin.

Measurements of porcine skin samples in Figure 7 show that the signature of the skin of the samples A, B, C, and D is observed through the six-layer gauze burn bandage and the light support bandage. The measurements also show that dressing materials increase the mean emissivity values by an amount ranging from 0.01 to 0.02. This increase is due to the increased in the losses in the dressing materials, this enhancing the MMW coupling of radiation to the skin through an impedance matching effect [50].

Measurements of porcine skin samples before and after the application of heat treatments in Figure 8 indicate that there is a clear signature for the burn that can be detected using the 90 GHz calibrated radiometer. The measurements indicate that the mean emissivity values of the burn-damaged skin are higher than that of unburned skin by 0.044, 0.084, 0.184, and 0.264 for samples X, Y, Z, and W and after 10, 60, 120, and 180 s of heat treatment applications respectively. This increase in the mean emissivity is due to the burning process that is removing the water from the skin, thereby reducing the reflectance and increasing the emissivity (η = 1 − R) [31]. The measurements also show that the signature of the burn is observed through the six-layers of gauze burn bandage and the light support bandage. These results indicate the transparency of the dressing materials over the MMW band [15].

Measurements of porcine skin samples before and after different applications of localized heat treatment on the same sample in Figure 9 indicate that different applications of heat treatments generate different burn depths. The differences in the mean emissivity values between unburned and burn-damaged skin are likely to increase with the degree of the burn, as emissivity is inversely proportional to the water content of the sample. These results indicate that radiometry might be used as a non-invasive (non-contact) technique to assess the degree of the burn.

Measurements of porcine skin sample in Figure 10 indicate that the mean emissivity value of the burned skin with exudates is lower than the unburned skin by ~0.09. Whereas, the mean emissivity values of the burned skin without exudates and with different burn depths are higher by ~0.07 and ~0.28. The interpretation of this is that burning process cauterizes the skin, resulting in exudates produced around the wound site. These exudates decrease the emissivity of the sample by an amount proportional to the amount of exudates produced. However, when all exudates evaporate because of the burning process, the sample emissivity reaches a maximum as there is almost no water remaining. After that, an extra application of heat treatment does not affect the emissivity of the sample, as illustrated in Figure 11. Future measurements on skin samples would include more variations in the sample types, looking perhaps at samples with fat layers, to investigate possible influence of this layer on the results.

The system depicted in Figure 3 represents a technology readiness level (TRL) three system; a proof-of-concept sensor for initial capability demonstrations. To progress the technology to the higher TRL’s where measurements could be made on patients would require appropriate redesign of the anechoic chamber and calibration procedure. Measurements with these follow-on demonstrators would be made with greater precision and convenience, offering a rapid and non-invasive (non-contact) diagnostic technique in a critical situation without necessity of dressing removal. Furthermore, the technique does not expose the patient or operator, to potentially damaging ionizing radiation so an easier technology to work with. More research is needed to establish other benefits.

## 5. Conclusions

The emissivity of porcine skin samples, with and without burns, was measured experimentally over the band of 80–100 GHz using radiometry. The measurements indicate the mean emissivity of burn-damaged skin without exudates is higher than unburned skin, whereas the mean emissivity of burn-damaged skin with exudates is lower than that of unburned skin. This means that the lower emissivities of burn-damaged skin are indicative of the presence of exudates, infection, and a non-healing state of the skin, whereas the higher emissivities are indicative of a dry burn, suggesting a full thickness burn (third-degree or deep second-degree burn). This indicates millimeter wave radiometry generates a clear signature from porcine skin burns and that this could be used on humans as a non-contact method to determine the severity of a burn in a matter of seconds.

These measurements confirm that the signatures of the burns can be measured through burn injury dressing materials (gauze burn dressing materials and light support bandage). This means that radiometry might be used as a non-contact technique for monitoring the healing status of the burn wounds under dressing materials. The measurements also indicate that radiometric sensitivity is sufficient to discriminate between the different burn depths, as the mean emissivities are directly related to the amount of localized heat treatment. This indicates that radiometry might be used as a non-contact technique to assess the burn depth or the degree of the burn.

## Figures and Tables

**Figure 1 sensors-19-02950-f001:**
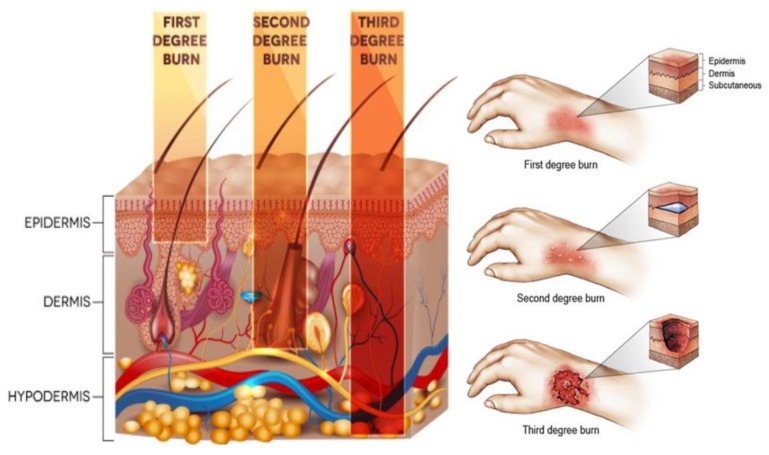
Illustration of different depth of invasion for burn injury [4,5].

**Figure 2 sensors-19-02950-f002:**
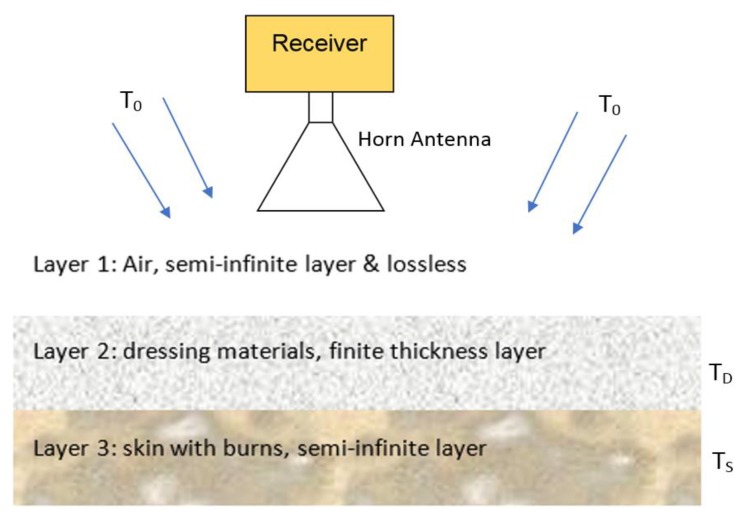
Three-layer model for dressed burn wound comprising of: a semi-infinite layer of air, finite thickness layer of dressing materials, and a semi-infinite layer of burn-damaged skin.

**Figure 3 sensors-19-02950-f003:**
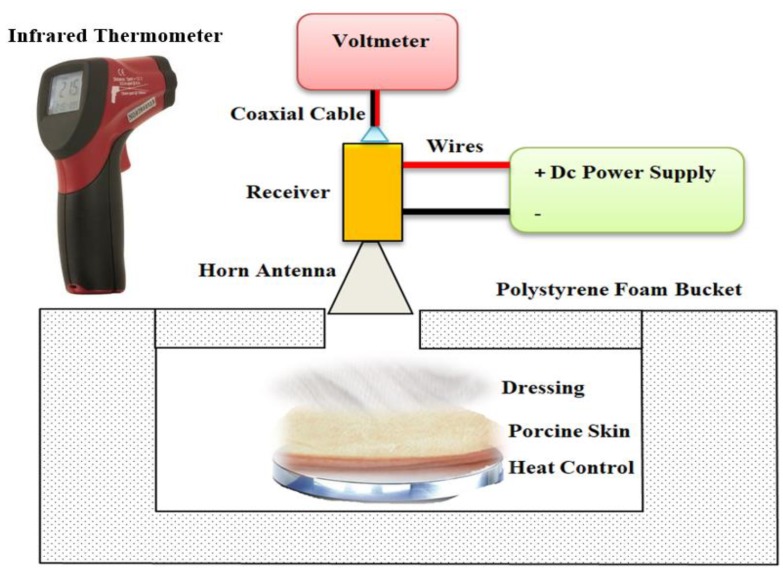
Experimental setup for the emissivity measurements of the porcine skin samples. A digital voltmeter is used to measure the output voltage level of the samples and a thermocouple and an infrared thermometer are used to measure the thermodynamic temperature of the samples.

**Figure 4 sensors-19-02950-f004:**
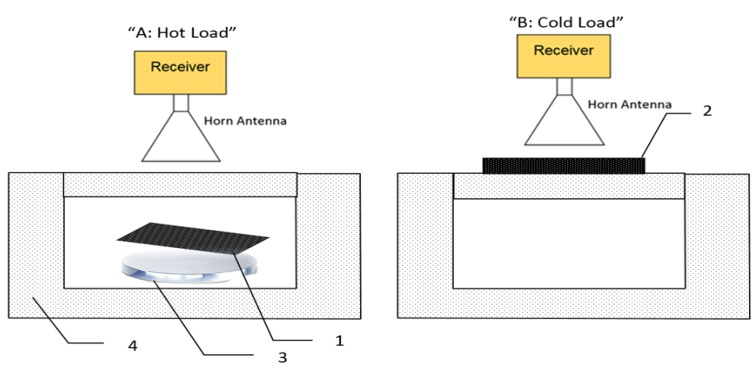
Radiometric emission centered at 90 GHz is collected by a moveable horn antenna at positions: A to measure a hot calibration source (**1**) (carbon loaded foam absorber; type: Eccosorb AN-73) stabilized at a temperature ~54 °C using a digital hotplate (**3**) placed in a polystyrene foam bucket (**4**), B to measure the cold calibration source (**2**) (carbon loaded foam absorber; type: Eccosorb AN-73) in thermodynamic equilibrium with air temperature ~20 °C.

**Figure 5 sensors-19-02950-f005:**
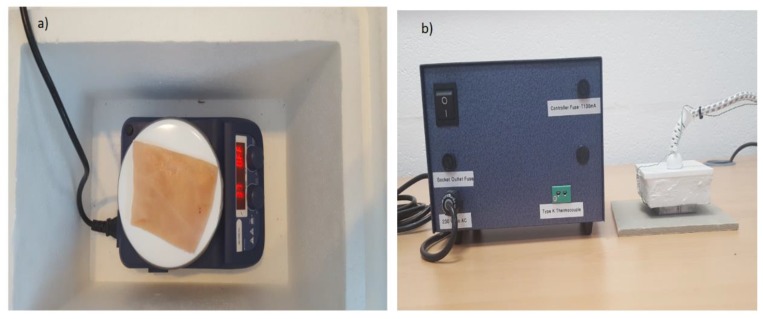
A digital hotplate used for heating and stabilizing the surface temperature of the porcine skin samples (**a**) and a heat control device with (50 × 50 mm) metal plate used for performing burns on the porcine skin samples (**b**).

**Figure 6 sensors-19-02950-f006:**
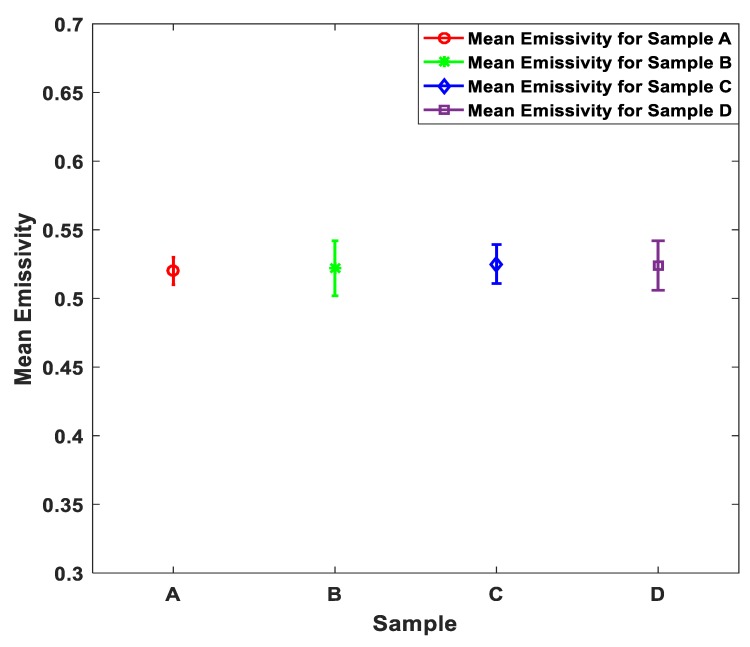
Mean emissivity values and standard deviation bars for porcine skin samples A, B, C, and D. The samples were taken from the back region of the same animal.

**Figure 7 sensors-19-02950-f007:**
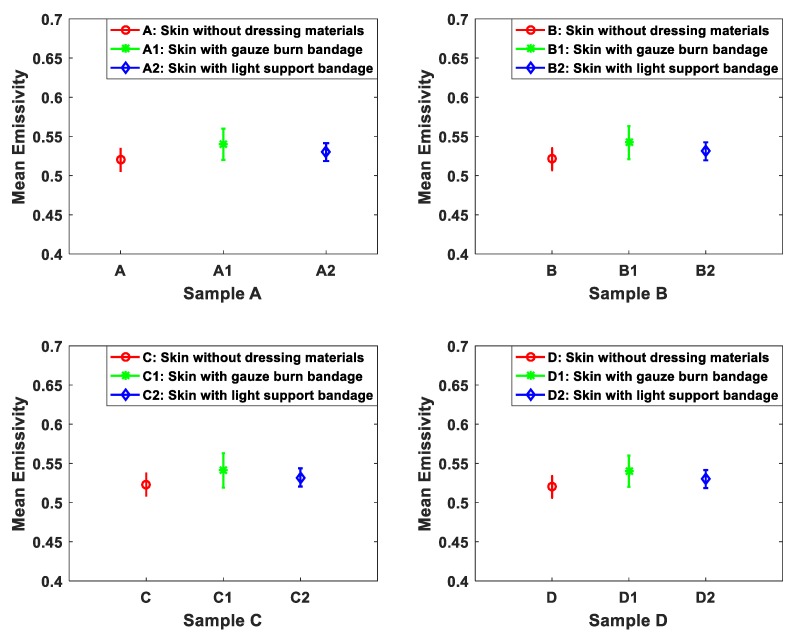
Mean emissivity values and standard deviation bars for porcine skin without and with dressing materials. The samples A, B, C, and D represent skin without dressing materials, A1, B1, C1, and D1 represent skin with six-layer gauze burn bandage, and A2, B2, C2, and D2 represent skin with a light support bandage.

**Figure 8 sensors-19-02950-f008:**
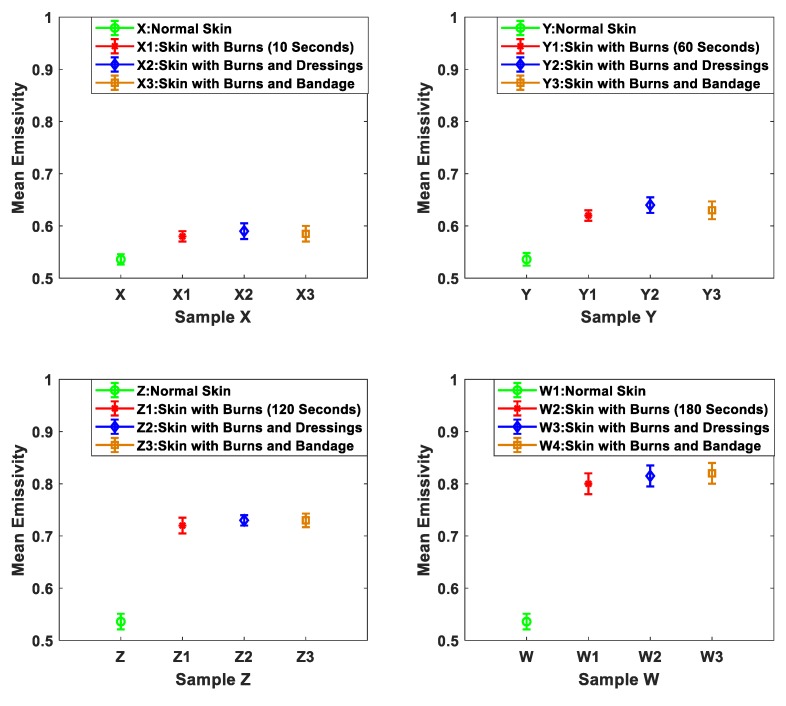
Mean emissivity values and standard deviation bars for porcine skin samples (obtained from the same animal) before and after the application of localized heat treatment. Samples X, Y, Z, and W represent normal skin. X1 represents skin with burns after 10 s of heat treatment, Y1 represent skin with burns after 60 s of heat treatment. Z1 represents skin with burns after 120 s of heat treatment, and W1 represents skin with burns after 180 s of heat treatment. X2, Y2, Z2, and W2 represent skin with burns and dressing materials (six gauze burn bandage). X3, Y3, Z3, and W3 represent skin with burns and single-layer light support bandage.

**Figure 9 sensors-19-02950-f009:**
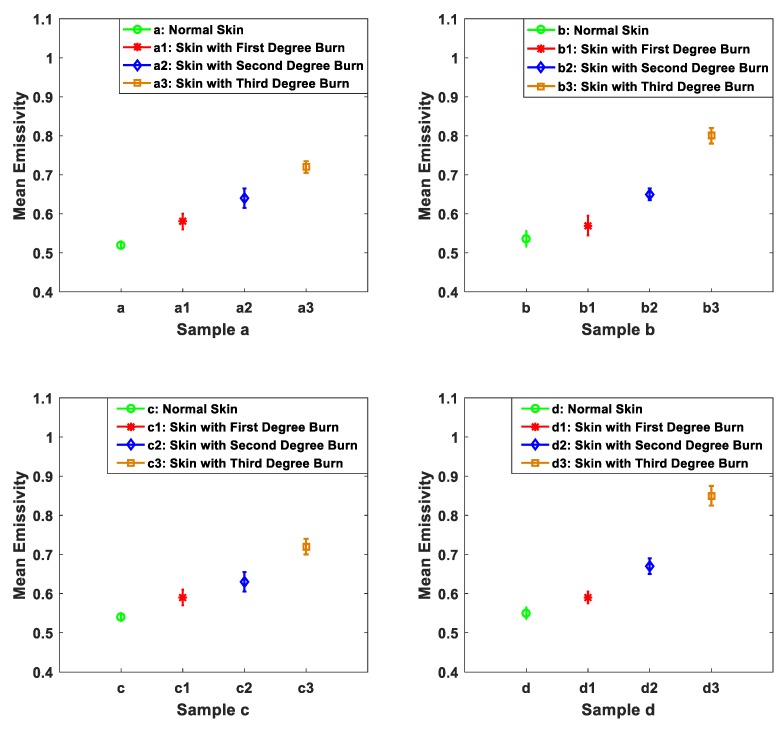
Mean emissivity values and standard deviation bars for porcine skin samples (obtained from different animals) before and after different applications of localized heat treatments. Samples a, b, c, and d represent normal skin; a1, b1, c1, and d1 represent skin with burns after 10 s of heat treatment (first degree burn); a2, b2, c2, and d2 represent skin with burns after 60 s of extra heat treatment (second degree burn); a3, b3, c3, and d3 represent skin with burns after 120 s of extra heat treatment (third degree burn).

**Figure 10 sensors-19-02950-f010:**
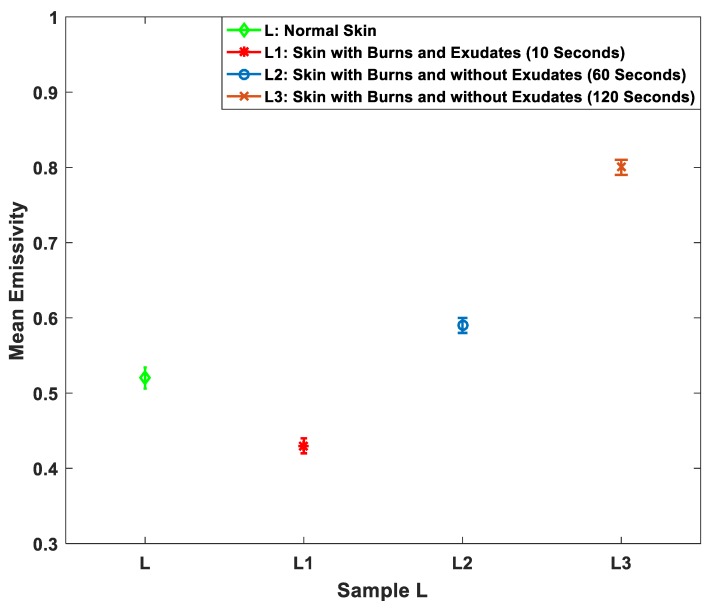
Mean emissivity values and standard deviation bars for porcine skin sample before and after different applications of localized heat treatments. L represents the normal skin; L1 represents skin with burns and exudates after 10 s of heat treatment, L2 represents skin with burns and without exudates after 60 s of extra heat treatment, L3 represents skin with burns and without exudates after 120 s of extra heat treatment.

**Figure 11 sensors-19-02950-f011:**
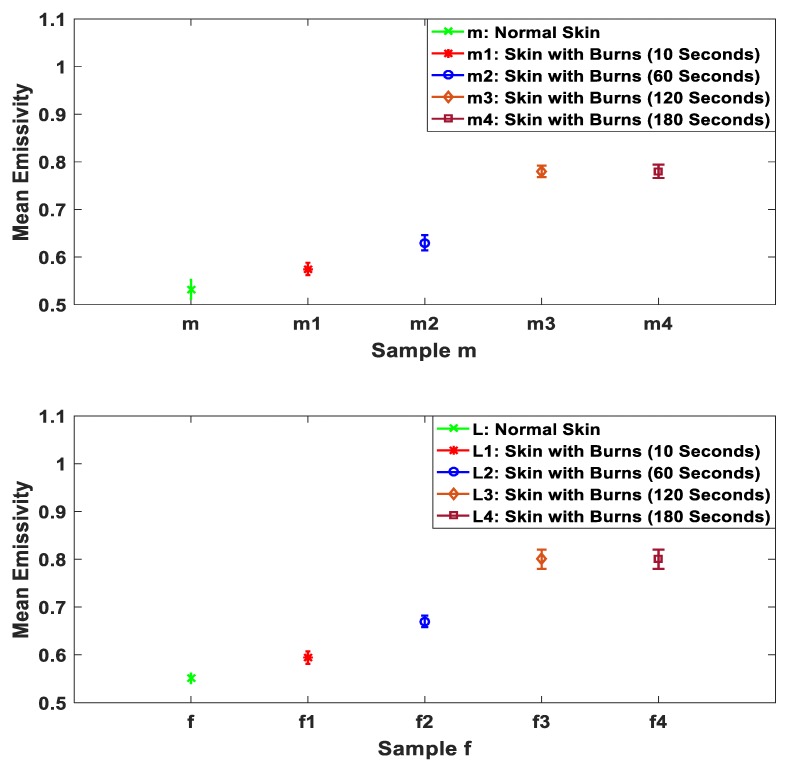
Mean emissivity values and standard deviation bars for porcine skin samples (obtained from different animals) before and after different applications of localized heat treatments. Samples m and f represent normal skin; m1 and f1 represent skin with burns after 10 s of localized heat treatment, m2 and f2 represent skin with burns after 60 s of extra heat treatment, m3 and f3 represent skin with burns after 120 s of extra application of heat treatment, and m4 and f4 represent skin with burns after 180 s of extra application of heat treatment.

**Table 1 sensors-19-02950-t001:** Overview of the main characteristics and features of the burn damaged skin.

Degree of Burn	Depth of Invasion	Signs	Degree of Pain
First degree or Superficial	Epidermis	Looks red without blisters and scars.	Painful
Second degree or Superficial dermal	Papillary dermis	Looks red with swelling, blisters, moisture, and scars.	Severe pain
Third degree or Deep dermal	Reticular dermis	Looks dry with pink and white color and severe scarring.	Pain and sensitivity are minimal
Fourth degree or Full thickness	The tissue under the skin (fat layer and beyond)	Looks dry with white, brown or black color and serious scarring.	No pain and no sensitivity because of the nerve ending destruction

**Table 2 sensors-19-02950-t002:** Mean emissivities for porcine skin after different applications of heat treatments.

Time Period for Heat Treatment (in Seconds)	Sample (a)	Sample (b)	Sample (c)	Sample (d)
0	0.52	0.536	0.54	0.55
10	0.58	0.57	0.59	0.59
60	0.64	0.65	0.63	0.67
120	0.72	0.8	0.72	0.85

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
