# Peer review of "Assessment of Bandaged Burn Wounds Using Porcine Skin and Millimetric Radiometry"

_sensors, 2019, doi:10.3390/s19132950_

Round 1

Reviewer 1 Report

Please explain the measurement system (electronic part) more explicitly. It is not clear what is being measured and how it is translated to a voltage in the output.

It is not clear how the presented method is realizable in real application since a complicated setup based on an anechoic medium is required.

The measurement is performed just on skin samples with no fat or tissue beneath it. In real cases the burn might reach to the tissue beneath the skin. Please explain how presence of a tissue affects your measurement results and method.

It is not clear what is the accuracy and advantages of the presented method over the current detection and measurement method and instruments.

Author Response

Many thanks for you constructive comments and suggestions. Please see the attachment.

Reviewer 2 Report

the paper proposes to use a radiometric system to measure the emissivity of the burned skin. This is an interesting method to know the degree of burning without having to remove the clothes.
From a scientific point of view, the study seems complete and rigorously conducted. This technique can have immediate benefits in the medical community.
I have no particular comments or suggestions.

Author Response

Many thanks for this. 

Reviewer 3 Report

This paper investigates the potential of using millimeter wave (MMW) radiometry as a new type of medical diagnostic to monitor burn wounds under dressing materials. The experimental measurements show that radiometry, as a non-contact method, is capable of distinguishing between normal and burn-damaged skin and assessing the burn depth or the degree of the burn under dressing materials without their often-painful removal.

The non-contact medical diagnostic to monitor burn wounds is novel and is an interesting and meaningful application. And there are some suggestions and questions as follows:

1. Please check the spelling of “Dessing” in the legend of Figure.8.

2. How to choose the distance of the radiometry to the skin and also the operating frequeny of the radiometry? Whether the power of radiometry cause some damage to the burt skin? Please give more details and explanations.

3. There are too many figures in the manuscript, it is better to combine the sub figures of different samples under the same condition, for example, Figures (a)-(d) can be combined to a single figure with a better comparison.

Author Response

Many thanks for your constructive suggestions and comments. Please see the attachment. 
